# EICIL: Joint Excitatory Inhibitory Cycle Iteration Learning for Deep Spiking Neural Networks

**Zihang Shao[1]**    **Xuanye Fang[1]**    **Yaxin Li[1]**    **Chaoran Feng[1]**
**Jiangrong Shen[2]**    **Qi Xu[1,2*]**

[1]School of Artificial Intelligence, Dalian University of Technology

[2]College of Computer Science and Technology, Zhejiang University

## Abstract

Spiking neural networks (SNNs) have undergone continuous development and extensive research over the decades to improve biological plausibility while optimizing energy efficiency. However, traditional deep SNN training methods have some limitations, and they rely on strategies such as pre-training and fine-tuning, indirect encoding and reconstruction, and approximate gradients. These strategies lack complete training models and lack biocompatibility. To overcome these limitations, we propose a novel learning method named Deep Spiking Neural Networks with Joint Excitatory Inhibition Loop Iterative Learning (EICIL). Inspired by biological neuron signal transmission, this method integrates excitatory and inhibitory behaviors in neurons, organically combining these two behavioral modes into one framework. EICIL significantly improves the biomimicry and adaptability of spiking neuron models and expands the representation space of spiking neurons. Extensive experiments based on EICIL and traditional learning methods show that EICIL outperforms traditional methods on various datasets such as CIFAR10 and CIFAR100, demonstrating the key role of learning methods that integrate both behaviors during training.

## 1 Introduction

In recent years, the spiking neural network (SNN) has garnered significant attention and research in the fields of neuroscience and artificial intelligence (AI) [33, 26, 7]. As a computational model that simulates the functioning principles of biological neural systems, SNN exhibits high biological plausibility and energy efficiency, making it a crucial approach for neural information processing[17, 20, 14]. However, traditional deep SNN training methods face certain limitations, such as reliance on pre-training and fine-tuning strategies, indirect encoding, and approximate gradients. These limitations lead to the training of SNNs leaning more towards deep learning-based methods [12, 17, 20, 3] and away from the principles of brain heuristics.

In order to enhance the biological plausibility and adaptability of SNN, and overcome these limitations, we propose a novel learning and training method called Excitatory-Inhibitory Cooperative Iterative Learning (EICIL). The core idea of EICIL is to simulate the excitatory and inhibitory behaviors of biological neurons and seamlessly integrate them into the training process of SNN. In this way, EICIL can better emulate the characteristics of neural signal transmission and expand the representation space of SNN.

---

[*] Corresponding author: xuqi@dlut.edu.cn

37th Conference on Neural Information Processing Systems (NeurIPS 2023).

In particular, EICIL uses an iterative training framework that emphasizes and optimizes the interaction between excitatory and inhibitory mechanisms. During the training process, EICIL combines excitatory and inhibitory mechanisms in both temporal and spatial dimensions, and enables collaborative training of the same network model structure. Additionally, we gradually adjust the ratio between excitatory and inhibitory mechanisms, aiming to approximate the ideal ratio of approximately 8:2[18]. This allows us to achieve better performance and effectiveness in SNN learning.

To validate the effectiveness of the EICIL method, we conducted extensive experiments and compared it with traditional deep SNN training methods. The experimental results demonstrate significant performance improvements of EICIL across multiple datasets, such as MNIST[5], CIFAR10 [19] and CIFAR100[20]. This further confirms the potential and advantages of the EICIL method in enhancing SNN learning capabilities. The main contributions in this paper are summarized as follows:

- This paper introduces a joint learning method based on excitatory and inhibitory mechanisms, namely the EICIL method. ELCIL provides a comprehensive and effective learning framework for deep SNN training and overcomes the limitations of traditional approaches. By integrating excitatory and inhibitory mechanisms in a cohesive manner, EICIL can better mimic the functional principles of the neural system, thereby enhancing the biological plausibility of SNN.

- The proposed method combines excitatory and inhibitory mechanisms for training. In addition, this paper introduces a multi-layer STDP (Spike-Timing-Dependent Plasticity) method as the inhibitory mechanism. At the same time, considering the poor training effect of the multi-layer STDP, we also modify and adjust the multi-layer STDP method. And keep it biologically interpretable and optimize its training results.

- EICIL has shown the ability to enhance the biological interpretability and adaptability of SNN, and expand its representation space. And it has achieved significant experimental results. This research is of great significance for further advancing the development of spiking neural networks in both theoretical and practical aspects.

## 2 Related Work

### 2.1 Learning of Spiking Neural Networks

In recent years, a number of learning methods have been developed to train deep SNNs with excellent performance, which can be categorized as ANN-to-SNN conversion (ANN2SNN) and end-to-end backpropagation methods. ANN2SNN pre-trains a source ANN and then converts it to an SNN by replacing the artificial neurons with spiking neurons. The main idea is to use the firing rate of the spiking neuron or average postsynaptic potential to approximate the ReLU activation of the artificial neuron. Although some advanced transformation methods achieve high accuracy on large-scale datasets in trained artificial neural networks with ResNet and VGG structures[6, 8, 16, 22], they ignore the rich temporal dynamics and require many time steps to achieve precoding precision. In contrast, backpropagation methods train SNNs directly by unfolding the network over time steps and then computing surrogate gradients in the spatial and temporal domains, which draws on the idea of training recurrent neural networks (RNNs) via temporal backpropagation [26, 31]. Since the gradient of the spikes with respect to the membrane potential is not differentiable, surrogate gradients are proposed by approximating the gradient with a smoothing function [25]. The backpropagation method with gradient surrogate can be applied to both static and neuromorphic datasets and requires much fewer time steps than the ANN2SNN method. Another backpropagation method is the time-based [32], which directly calculates the gradient from the spike firing time to the membrane potential without expanding the network over the time step. Inspired by the residual connections of Resnet and VGG network models, our EICIL improves the connection structure of residual networks to increase the representation space of SNN network models and improve the performance of SNNs.

### 2.2 Biological interpretability of Spiking Neural Networks

The Surrogate Gradient Method(GS)[1, 27] and Spike-Timing-Dependent Plasticity(STDP)[4, 29] are two concepts commonly used in neuroscience to describe the synchronization behavior between neurons and synaptic plasticity, which can be used for weight learning in convolutional and fully

connected layers in SNNs. However, GS is too simplified for the synchronization relationship between neurons, but synchronization in the brain is relatively complex, and there exist different forms such as partial synchronization and phase synchronization, so there are limitations in explaining and modeling the real synchronization phenomenon in the brain [24, 28, 30, 23]. Although STDP has great significance in describing synaptic plasticity, STDP ignores other forms of synaptic plasticity [2], such as long-range synaptic enhancement (LTP) and long-range synaptic inhibition (LTD).In order to make up for the shortcomings of the above methods, based on the idea of spike/no-spike classification[10, 13, 15] and combining the characteristics of STDP and GS, we developed an iterative training method of STDP and GS, aiming to improve the expression ability of spiking neurons and SNN.

## 3 Preliminary

### 3.1 Traditional IF

Integrate Fire(IF)[11]is based on the principle of integration and firing of neurons for information transfer. Neurons receive input spike sequences from Presynaptic neurons and integrate the input signals by weighting and summing the input spike sequences. The weights used in the integration process can be adjusted according to the specific task and network structure. When the integrated potential reaches the threshold of the neuron, the neuron fires a spike. At this point, the integrating potential resets to the baseline level, and the neuron undergoes a period of absolute underdrive. It does not respond to any input spikes during this time.The IF model is calculated as Eq. (4)

$$V(t) = \sum_i w_i \cdot S(t - t_i) \tag{1}$$

where $V(t)$ denotes the integrated potential of the neuron at time $t$, $w_i$ denotes the weight of the $i$th input spike, and $S(t - t_i)$ is the firing function that represents the effect of the input spike at the moment of time $t - t_i$.

The condition for determining whether a neuron issues an output spike is

$$V(t) \geq V_{\text{threshold}} \tag{2}$$

If the integrated potential $V(t)$ exceeds the threshold $V_{\text{threshold}}$, the neuron will fire a spike.

### 3.2 Excitatory and inhibitory in neural systems

In the biological nervous system, signal transmission models mainly include two aspects: synaptic transmission and intra-neuronal electrical signal transmission. In terms of synaptic transmission, there are mainly two types: excitatory transmission and inhibitory transmission. Excitatory and inhibitory neurons are usually combined to produce complex behaviors and functions. For example, in the cerebral cortex, there is a widespread neural network between excitatory and inhibitory neurons, which can support higher functions such as perception, cognition, and decision-making. In this type of network, excitatory and inhibitory neurons are usually mixed in a certain proportion. For example, in the visual cortex, about 80 percent of neurons are excitatory neurons, and 20 percent are inhibitory neurons. This proportion may vary in different brain regions and different biological species. In this combination, excitatory neurons can activate adjacent neurons by generating spiking signals, while inhibitory neurons can suppress adjacent neurons by generating opposite spiking signals. This interaction between excitatory and inhibitory can generate complex dynamic behaviors and support information processing and control functions in the nervous system.

## 4 Method

### 4.1 Surrogate Gradient Method

The Surrogate Gradient Method is a commonly used technique in training spiking neural networks (SNNs) that replaces continuous functions in gradient descent with surrogate gradient functions to generate neural spiking output. In the backpropagation algorithm of SNNs, the gradient of each neuron is discrete in time, but traditional gradient descent algorithms require continuous gradient

values to update weights, which necessitates converting discrete gradient values into continuous functions. GS achieves the use of discrete gradients by replacing these continuous functions with spike functions[1]. In this study, we select GS as the excitatory signal transmission method and choose sigmoid function as the gradient surrogate function. At the same time, we use the most basic neuron model to generate spikes. Below is the description of the sigmoid gradient approximation function: sigmoid function itself is defined as:

$$sigmoid(x) = \frac{1}{1 + e^{-x}} \tag{3}$$

$$sigmoid\_grad(x) = sigmoid(x) \cdot (1 - sigmoid(x)) \tag{4}$$

### 4.2 Multi-STDP

STDP is a synaptic plasticity rule based on spiking neurons, which simulates the process of synapses continuously strengthening or weakening with the interaction between neurons.When a neuronal synapse receives a series of spikes, there will be plastic changes due to the difference in the timing of the firing of the spikes between the pre- and post-synaptic neurons. Specifically, if the pre-synaptic neuron spikes before the post-synaptic neuron, the strength of the synapse increases, while if the pre-synaptic neuron spikes after the post-synaptic neuron, the strength of the synapse decreases. Let $w_{ij}(t)$ be the weight of the synapse from neuron $i$ to neuron $j$ at time $t$, and $\Delta t = t_j - t_i$ be the time difference between the spikes of neurons $i$ and $j$. The STDP rule can be expressed as Eq. (5)

$$\begin{cases} \Delta w_{ij}(t) = \eta_+ \cdot f_+(\Delta t) \cdot (1 - w_{ij}(t)), & \Delta t > 0 \\ \Delta w_{ij}(t) = -\eta_- \cdot f_-(\Delta t) \cdot w_{ij}(t), & \Delta t < 0 \\ \Delta w_{ij}(t) = 0, & \Delta t = 0 \end{cases} \tag{5}$$

Where, $\eta_+$ and $\eta_-$ represent the learning rates for synaptic potentiation and depression, respectively, and $f_+$ and $f_-$ denote the weight update functions for positive and negative spike timing differences. The commonly used weight change function is exponential.

$$\begin{cases} f_+(\Delta t) = e^{-\Delta t/\tau_+} \\ f_-(\Delta t) = e^{\Delta t/\tau_-} \end{cases} \tag{6}$$

Where, $\tau_+$ and $\tau_-$ are the time constants for synaptic potentiation and depression, respectively.

Although for many years, due to factors such as the dynamic nature of neurons, complex inter-neuronal connections, and training time and computational costs, multi-layer STDP models have been rarely used by researchers[21]. However, in terms of lateral inhibition, the STDP rule can make the lateral inhibition effect of a neuron more significant. Therefore, we adopt a multi-layer STDP model as the inhibitory mechanism of the EICIL structure.

In a multi-layer STDP model, the STDP method is utilized to perform temporal logic processing of spike neuron propagation. In this model, the STDP rule is employed for weight adjustments in the linear layer, and convolutional layer.

The STDP rule adjusts connection weights based on timing differences in firing spikes between neurons. When the timing of the spikes between neurons is consistent, the connection weight is strengthened. When the timing difference is large, the connection weight is weakened. So the strength of connections between neurons is adjusted according to their activity patterns. In this way, the connection weights between neurons in the multi-layer STDP model will be adjusted according to their temporal relationship, thereby achieving multi-layer propagation. This timing-dependent plasticity mechanism allows the neural network to adaptively adjust the connection strength during the learning process to suit the statistical characteristics of the input data and task requirements.

In the initial experiments, we found that the original multi-layer STDP method performed extremely poorly in training across three network structures. It even failed to reach an accuracy of 50 percent on the CIFAR10 dataset. Upon analyzing the multi-layer STDP model and its code, we discovered that the activation of neurons and synaptic weight updates only occurred during the forward propagation process. This limitation meant that the gradient information from backpropagation could not directly propagate to earlier layers, thereby restricting the model's learning capability and training effectiveness. Additionally, the inability to effectively transmit gradient information to earlier layers resulted in either diminishing gradients (gradient vanishing) or exponentially increasing gradients

(gradient explosion). This instability during the training process made it challenging to achieve good convergence. Furthermore, the model lacked the ability to simultaneously adjust weights across the entire network to minimize the overall loss function. Therefore, after careful consideration, we introduced the STDP-BW method by incorporating the backpropagation technique into the STDP model. The integration of the STDP model with backpropagation presents a comprehensive and enhanced framework for training multi-layer networks. This approach combines the biological principles of synaptic plasticity with the optimization capabilities of backpropagation. It not only takes into account the biological principles of synaptic plasticity but also leverages the optimization power of backpropagation. This integration provides a more comprehensive and advanced framework for training multi-layer networks.

## 4.3 The combination of excitability and inhibition

### 4.3.1 Approach I: Heterogeneous Learning

We combine excitatory and inhibitory mechanisms by using them to compute different types of network layers within the same network structure. The excitatory mechanism is represented by the gradient replacement method, and the inhibitory mechanism is represented by the STDP method. The neural models for the gradient replacement method and the STDP method are denoted by $F(\cdot)$ and $G(\cdot)$, respectively. For a neuron $j$, its membrane potential $V_j(t)$ at time $t$ changes according to:

$$\frac{dV_j(t)}{dt} = \sum_{i=1}^{N} w_{ij} F(s_j(t - \Delta) - \theta_j) + \sum_{i=1}^{N} w_{ij} G(s_j(t - \Delta) - s_i(t - \Delta) - \theta_{ij}) \qquad (7)$$

where $w_{ij}$ represents the connection weight between neurons $i$ and $j$, $\theta_j$ represents the threshold of neuron $j$, $s_j(t)$ represents the signal indicating whether neuron $j$ has fired at time $t$, and $\Delta$ represents the time delay of spiking propagation. This equation expresses the rate of change in the membrane potential of a neuron in response to input from other neurons, representing its excitatory or inhibitory nature.

In this study, we propose a novel approach combining STDP models, backpropagation, etc. to train different layers of a neural network. We utilize the STDP model to train the last linear layer in the network to dynamically update the weights according to the spiking neural network. For the convolutional layer and other layers, we employ a gradient-based method to optimize the weights. This framework facilitates the integration of excitatory and inhibitory training by applying STDP-based weight adjustments across the network after each training round.

Furthermore, we conducted additional experiments to explore various training configurations. First, we exclusively trained the convolutional layer using the STDP model while utilizing the gradient-based method to train the remaining layers. This allowed us to evaluate the individual impact of STDP training on the convolutional layer. Additionally, we investigated a hybrid approach where the convolutional layer was trained using both STDP and backpropagation, while the other layers were trained using the gradient-based method. This multiple-STDP-based inhibitory training regime enabled iterative fine-tuning of the neural network's weight parameters, creating a dynamic interplay between excitatory and inhibitory processes. We refer to this method as STDP-BW-GS.

### 4.3.2 Approach II: Cycle Iteration Learning

The second method involves training different epochs of the same network using both excitatory and inhibitory mechanisms within the network's structural framework. This approach entails training the complete network with different methods at distinct time steps, effectively encompassing a single training session within each time interval.

In terms of biological interpretability, excitatory and inhibitory mechanisms exhibit alternating and interacting patterns within neurons, employing diverse mechanisms to influence signal propagation. In the biological neural system, signal transmission between neurons is subject to temporal dynamics, including time delays and other temporal information. The integration and analysis of these temporally modulated signals contribute to processing the input information before transmitting it for further analysis in the brain. Motivated by this, we adopt distinct methods to train neurons during different epochs. Each epoch represents a training cycle within the network model, utilizing different time steps as intervals to train the network using excitatory and inhibitory mechanisms. We

leverage the current optimal results obtained from the excitatory mechanism to train the inhibitory mechanism, and reciprocally, use the current optimal results from the inhibitory mechanism to train the excitatory mechanism. This iterative process facilitates the network model's adaptation to both excitatory and inhibitory mechanisms, culminating in the attainment of optimal model parameters through synergistic training.

In this case, the network's connection weights at each time step can be represented as $W_t$, where $t$ denotes the time step. At each time step $t$, the following update rules using gradient replacement and STDP can be employed as Eq. (8) and Eq. (9):

$$W_{t+1} = W_t - \eta \frac{\partial W_t}{\partial L} \tag{8}$$

$$\Delta W_{ij}(t) = \sum_{k \in S^+} f_+(t - t_k) - \sum_{k \in S^-} f_-(t - t_k) \tag{9}$$

Where, $L$ represents the network's loss function, $\eta$ denotes the learning rate, and $i$ and $j$ represent the indices of the neurons. $S^+$ and $S^-$ correspond to the excitatory and inhibitory spiking sequences received by neuron $j$ from neuron $i$, respectively, and $f_+$ and $f_-$ represent the spatiotemporal weight functions for excitatory and inhibitory spiking. The updated connection weights can be obtained as Eq. (10)

$$W_{t+1} = W_t + \eta \Delta W_t \tag{10}$$

This approach can preserve the biological realism of SNN models to a certain extent while harnessing the powerful capabilities of deep learning.

Simultaneously, considering that the excitatory and inhibitory mechanisms in the visual neural network of the human brain do not exhibit a 1:1 ratio, we introduce a proportional adjustment of training durations for these mechanisms by controlling the ratio of time steps. Specifically, we introduce a parameter K to modulate the relative durations of training for the excitatory and inhibitory mechanisms. When K is a positive value, it indicates that the inhibitory mechanism accounts for a proportion of 1/K. Conversely, when K is a negative value, it indicates that the excitatory mechanism accounts for a proportion of -1/K.By tuning the value of K, we seek to determine the optimal ratio. Here, we refer to the iteration using the GS method and STDP-BW method as GSI, and the iteration using the GS method and STDP-BW-GS method as GSGI.

## 4.4 Network model based on excitatory inhibition

Combined with the fusion method in 4.3.1, the proportion of excitatory mechanism and inhibitory mechanisms in the biological neural network is adjusted at the same time. We propose an idea: build a network model, set some layers as inhibitory mechanisms and some layers as excitatory mechanisms, and train the two mechanisms in this network model. At the same time, the ratio of excitatory and inhibitory mechanisms in this network model is also 4:1[18], that is, the excitatory mechanism accounts for about 80%, and the inhibitory mechanism accounts for about 20%. The method of the inhibitory mechanism we choose in this paper is the STDP method, and the STDP method has some limitations: it can only perform temporal logic training on the convolutional layer and the linear layer. Since the STDP method is used to train the last linear layer of the network structure separately, the structure of the network model ensures that the ratio of excitatory and inhibitory mechanisms reaches an ideal ratio is small. So it is not necessarily possible to train to produce the best results. We use the STDP method to train the convolutional layer and combine the framework of the Resnet network model to build our own network model. The model consists of basic convolutional layers, BN layers, pooling layers, residual blocks, and linear layers. Each residual block contains two sets of convolutional layers, maximum pooling layers, BN layers, and spike neurons. We train the convolutional layers with an inhibitory mechanism, and the linear, pooling, and BN layers with an excitatory mechanism. Finally, the constructed model reached 77% of the excitatory mechanism and 23% of the inhibitory mechanism, which is close to the ratio of excitation and inhibition in the human neural network.

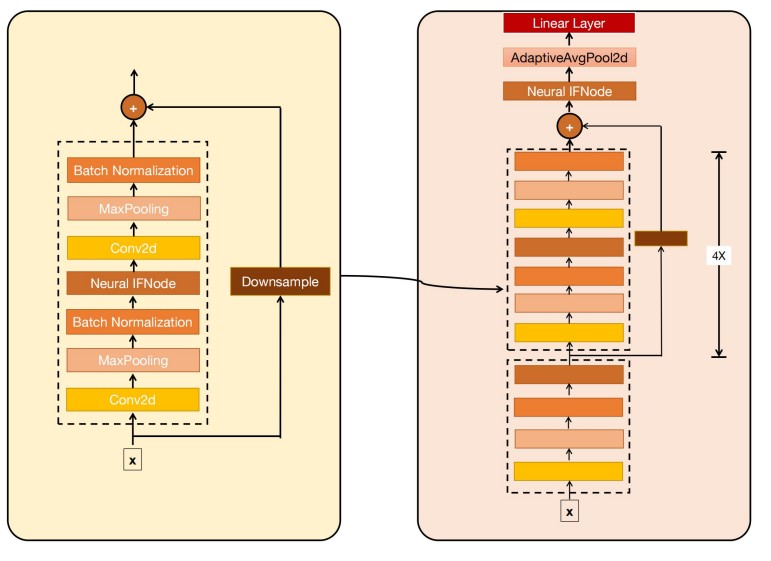

Figure 1: The figure above presents the framework of the excitatory inhibition network. On the left side, we can observe the residual blocks of the network, while on the right side, the overall network architecture is depicted. The design of the network framework ensures a ratio of 8:2 between excitatory and inhibitory mechanisms.

# 5 Experiment

In this section, we demonstrate the effectiveness of our proposed EICIL model. We first compare our results with other existing state-of-the-art, then carry out ablation studies to evaluate different aspects of our proposed method.

## 5.1 Experimental Settings

We conduct experiments on a server equipped with a 16-core Intel(R) Xeon(R) Xeon(R) Gold 6330 2.80GHz CPU and 20 NVidia GeForce RTX 3090 Ti GPUs. The training of SNN is based on the spikingjelly framework [9]. Our experiments mainly use the three network frameworks of Resnet, VGG, and the network we built, and conduct experiments on three image data sets of MNIST[5], CIFAR10 [19] and CIFAR100[20]. In this experiment, we adopt different training methods for the same experimental framework to better verify the training advantages of iterative learning.

**MNIST.**MNIST is a picture dataset of handwritten digits. It counts pictures of handwritten digits from 250 different people. The dataset is relatively simple and can initially demonstrate the performance of our model. When training, we use Resnet and VGG11 for training to get the results of different training methods.

**CIFAR 10 & CIFAR 100.** The CIFAR dataset was collected from 80 million RGB miniature image datasets. The CIFAR dataset can be divided into CIFAR-10 and CIFAR-100 according to the number of classified objects involved. The dataset is relatively complex and can more realistically verify the performance of our proposed framework. When training, we need to first encode the static images into spike sequences, and then input them into the SNN, here the first spike neuron layer in the network is used as the encoding layer.

## 5.2 Evaluation on STDP-BW Method

Experimental results have demonstrated that the incorporation of backpropagation into the STDP model (STDP-BW) achieves an accuracy of 90.39 percent the CIFAR10 dataset, which is a 1percent

Table 1: Test accuracies of EICIL on CIFAR 10, CIFAR 100, and MNIST.

| SNN Model | Database | GS (Acc. %) | STDP-BW (Acc. %) | STDP-BW-GS (Acc. %) | GSI (Acc. %) | GSGI (Acc. %) | Improvment (Acc. %) |
|---|---|---|---|---|---|---|---|
| **Spiking Resnet 18[17]** | CIFAR10 | 89.39 | 90.39 | 89.16 | 90.05 | 90.34 | 0.95 |
| | CIFAR100 | 56.28 | 56.20 | 57.37 | 63.03 | 63.47 | 7.19 |
| | MNIST | 99.17 | 99.35 | 99.26 | 99.36 | 99.44 | 0.27 |
| **Spiking VGG 11[22]** | CIFAR10 | 83.96 | 84.06 | 83.96 | 84.11 | 85.31 | 1.35 |
| | CIFAR100 | 54.01 | 54.63 | 52.44 | 55.44 | 55.71 | 1.70 |
| | MNIST | 98.2 | 99.09 | 95.85 | 98.55 | 98.77 | 0.55 |
| **Excitatory inhibition Net** | CIFAR10 | 88.88 | 89.14 | 89.64 | 89.23 | 89.43 | 0.55 |
| | CIFAR100 | 53.61 | 53.92 | 54.03 | 54.61 | 53.86 | 0.25 |
| | MNIST | 99.33 | 99.34 | 99.30 | 99.33 | 99.35 | 0.02 |

improvement compared to the GS method. Comparable accuracy was observed on the CIFAR100 and MNIST datasets when compared to the GS method. Furthermore, a comparison with other network models such as Spiking Resnet18, Spiking VGG11, and Excitatory Inhibition Net reveals that the Spiking Resnet18 model outperforms the other two in terms of performance.

## 5.3 Evaluation on Iteration-based Methods

In the experiment, we continue to use Spiking Resnet18, Spiking VGG11 and the spiking form of our proposed Excitatory inhibitor Net to test the two training methods of GSI and GSGI, and then analyze the model.

For the MNIST data set, as shown in Table 1, when using Spiking Resnet18, GSI is improved by 0.19% compared to the original GS method, GSGI is improved by 0.28%, when using Spiking VGG 11, GSI is compared to the original The GS method has been improved by 0.35%, and the GSGI has been improved by 0.57%, which shows that the iterative method can improve the expression ability of the enhanced neuron model. To better compare the effectiveness of using an iterative training approach, we visualize a line graph of test accuracy.

For the CIFAR10 data set, when Table 1 uses Spiking Resnet18, GSI improves by 0.66% compared to the original GS method, and GSGI further improves by 0.95%, while the model performance of VGG11 is similar. The improvement is 0.15% and 1.35%, respectively, compared with the MNIST dataset, the algorithm based on iteration performs better on larger datasets.

For the CIFAR100 dataset, as shown in Table 1, when using Spiking Resnet18, GSI has improved by 6.75% compared to the original GS method, and GSGI has further improved by 7.19%, but due to the model structure and complexity The reason for this is that the performance of the VGG11 model is relatively poor, with only an increase of 1.43% and 1.70% respectively. The algorithm based on iteration performs better on larger data sets, which to a certain extent shows that the method based on iteration can improve the network's Biological interpretability, which can more effectively use the excitatory and inhibitory information of neurons for collaborative training.

For our Excitatory inhibition Network, as shown in Table 1, the performance of the GSI and GSGI iterative models is slightly better than that of Spiking VGG but slightly lower than that of Spiking Resnet 18. This shows that the iterative method has a stronger matching ability with the Resnet network structure and can greatly improve its expression. Ability, and model accuracy in the MNIST, CIFAR10, CIFAR100 data sets all show a positive improvement effect.

At the same time, comparing the GSGI and GSI results of the three data sets in the three networks, the accuracy of GSGI is higher than that of GSI, which also proves that the STDP-BW-GS method is slightly better than the STDP-BW method. It shows that the combination of excitatory mechanism and inhibitory mechanism proposed by us will lead to the improvement of training accuracy.

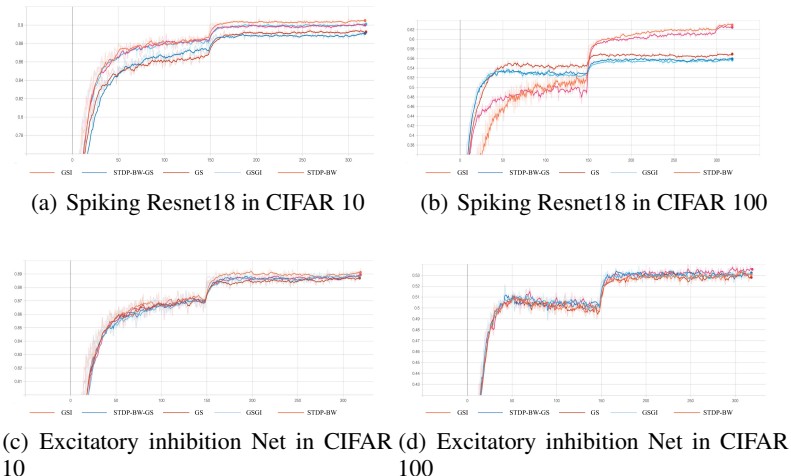

(a) Spiking Resnet18 in CIFAR 10      (b) Spiking Resnet18 in CIFAR 100

(c) Excitatory inhibition Net in CIFAR 10      (d) Excitatory inhibition Net in CIFAR 100

Figure 2: Performance of test accuracy of Spiking Resnet 18 and Excitatory inhibition Net in CIFAR 10 and CIFAR 100

Table 2: Test accuracies of EICIL on CIFAR 10, CIFAR 100, and MNIST.

| SNN Model | Parameter k | Learning Method | MNIST (Acc.%) | CIFAR 10 Acc. (%) | CIFAR 100 Acc. (%) |
|---|---|---|---|---|---|
| Spiking Resnet 18[17] | 3 | GSI | 99.39 | 89.97 | 57.74 |
| | | GSGI | 99.42 | 90.26 | 62.24 |
| | 4 | GSI | 99.33 | 90.16 | 62.59 |
| | | GSGI | 99.43 | 90.21 | 62.64 |
| | 5 | GSI | 99.30 | 90.46 | 56.15 |
| | | GSGI | 99.34 | 90.58 | 66.36 |
| | -3 | GSI | 99.36 | 89.97 | 57.11 |
| | | GSGI | 99.37 | 89.56 | 64.28 |
| | -4 | GSI | 99.34 | 90.38 | 58.53 |
| | | GSGI | 99.36 | 90.55 | 63.16 |
| | -5 | GSI | 99.32 | 90.34 | 56.62 |
| | | GSGI | 99.34 | 90.01 | 62.86 |

## 5.4 Ablation Study

**The effect of the hyperparameter k on the expressiveness of the model.**

Considering that different k values have different effects on the model, we conducted ablation experiments and tested the iterative training models GSI and GSGI based on Spiking Resnet 18. The experimental results are shown in Table.2, in the MNIST dataset On the above, the accuracy difference between GSI and GSGI ranges from 0.03% to 0.10%. On the CIFAR10 data set, the accuracy difference between GSI and GSGI ranges from -0.33% to 0.29%. On the CIFAR100 data On the set, the accuracy difference between GSI and GSGI ranges from 0.05% to 8.62%. We found that, limited by the size of the data set, the accuracy improvement on the MNIST and CIFAR10 data sets is small, while under the CIFAR100 large data set, the expressive ability of the model is significantly improved. At the same time, we found that the larger the K value, the higher the accuracy rate and the better the excitation-inhibition trade-off ability. When the K value reaches 5, the ideal 8:2 ratio of excitation and inhibition of the visual nervous system in the biological brain has been reached, which happens to be the highest accuracy rate of GSI and GSGI at this time. Therefore, it is also proved that our model method is more biologically interpretable while improving accuracy.

# 6  Conclusion

Inspired by the idea of neuron excitation-inhibition, this paper proposes a joint learning method based on excitation and inhibition mechanisms. The main idea is to better simulate the functional principles of the nervous system by integrating excitation and inhibition mechanisms in a cohesive manner. Thereby enhancing the biological rationality of SNN and improving the expressive ability and accuracy of spiking neural networks.

Considering the limited expressive power of the model structure, the experimental results show that we can extend the work to more situations and improve the accuracy of the model under specific tasks through iterative training. It also shows that the proposed model and training method exhibit good performance on both small and relatively large datasets.

In our future work, we will extend the joint learning method to exploit the biological properties of excitability and inhibition, continue to explore how the two are combined and paired, and evaluate them on more datasets.

# 7  Acknowledgement

This work was supported in part by National Natural Science Foundation of China (NSFC No.62206037,62306274), National Key Research and Development Program of China (2021ZD0109803), the China Postdoctoral Science Foundation under Grant No. 2023M733067 and No. 2023T160567, and the Fundamental Research Funds for the Central Universities (DUT21RC(3)091)

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
