# OpenReview forum: "EICIL: Joint Excitatory Inhibitory Cycle Iteration Learning for Deep Spiking Neural Networks"
_NeurIPS.cc/2023/Conference — NeurIPS 2023 poster_

### Official Review · Reviewer_5jTx · 2023-06-13

**Soundness:** 3 good
**Presentation:** 2 fair
**Contribution:** 3 good
**Rating:** 7
**Confidence:** 5

**Summary:**

The authors introduce a novel training method for spiking neural networks which combines both excitatory and inhibitory mechanisms for iterative training. This method uses the time step of spiking neural networks to train the same network with two training methods in the overall training process.At the same time ,the authors design the training time ratio of the two, reaching the ideal ratio of 8:2 of excitatory and inhibitory mechanisms in the nervous system of biological brains. It enhances the biological interpretability of spiking neural networks.

**Strengths:**

1:The training method that combines excitatory and inhibitory mechanisms is an innovative breakthrough for spiking neural networks, making spiking neural networks more biologically interpretable.
2:The new training idea proposed in this paper expands the training thought of spiking neural networks. The excitatory and inhibitory mechanisms use different time to train the same network structure in one overall training process, making full use of the time characteristics of spiking neural networks.
3:The paper also designs the ideal ratio of 8:2 of excitatory and inhibitory mechanisms to the neural network model in biological brains, making the above method more persuasive.

**Weaknesses:**

1:The author’s writing is immature and the expression is not clear enough.
2:The paper only introduces the combination of one excitatory mechanism and one inhibitory mechanism, and does not fully express the effectiveness and feasibility of the method.
3:The methods and neurons used are basic and do not fully reflect the improvement of the current highest accuracy.

**Questions:**

1. The paper only introduces one combination of excitatory and inhibitory mechanisms. Can you add another combination of excitatory and inhibitory mechanisms to fully demonstrate the feasibility of this training method?
2. The datasets used in the paper are some more traditional datasets. I wonder if they can have the same effect on new datasets.
3. Compared with 1:1 training, the positive and negative training results of K value have improved. Is this the effect of excitatory and inhibitory methods, or the result of 8:2 or 2:8 ratio of excitatory and inhibitory in spiking neural networks from the biological perspective of 8:2 ratio?

**Limitations:**

As I mentioned in weaknesses,the methods and neurons used in the paper are the most basic, using more advanced methods or neurons may be a limitation that can be solved in the future.Overall I see no negative societal impact to discuss.

---

> ### Author Rebuttal · Authors · 2023-08-08
>
> Thank you very much for your careful review and constructive comments, which have much contributed to improving the paper. We will address all the comments in the paper and provide responses to the comments below.
>
> We have made revisions for writing problems
>
> For Question1:
>
> For the applicability of another combination, the iterative training method that we propose combines excitation and inhibition is aimed at training the same network with the excitatory mechanism and the inhibitory mechanism. In theory, as long as the two methods can be used in the same network If the training is carried out in the middle, then no matter which training method is selected as the excitatory mechanism or the inhibitory mechanism, as long as one method is the excitatory mechanism and the other is the inhibitory mechanism, then the method is effective. At the same time, we consider that the two existing methods are sufficiently representative in the training methods of spiking neural networks, and due to equipment limitations, there is not enough time for experiments. But in the future, we will explore what method is more suitable for our method, and continue to explore how to optimize the method and improve the efficiency and accuracy of the combination of the two.
>
> For Question2:
>
> As for whether our method proposed by you can be applied to other more complex data sets, we have added a set of experiments. The experimental data set is the DVSGesture data set. The "DVSGesture" data set is a gesture recognition data captured based on dynamic visual sensors set. The main feature of the DVSGesture dataset is that it contains impulsive event sequences of gesture motions captured on dynamic vision sensors. Each pulse event records a change in pixel value within the region of interest, usually expressed as a timestamp, pixel coordinate, and polarity (increase or decrease).Please refer to Table 3(See the attachment or the table below for details) for the experimental results. The experimental results show that the STDP-BW-GS method reaches 95.14\%, which is higher than the 94.79\% of the STDP-BW method, which confirms the idea of method 4.3.1. At the same time, the results of GSGI and GSI are both higher than the accuracy of the GS method baseline by at least 2\%, which proves the effectiveness of the idea in method4.3.2. And the experimental results also show that the result of GSGI is higher than that of GSI, which also proves that the STDP-BW-GS method is better than STDP-BW. Therefore, experiments have proved that our method is also applicable to more complex data sets, and the robustness and generalization of the method have been verified.The table is shown below：
>                                               Table3:Test accuracies of EICIL on DVS Gesture.
> _____________________________________________________________________________________________________________
> | SNN Model &nbsp;&nbsp;&nbsp;&nbsp;&nbsp;&nbsp;&nbsp;&nbsp;&nbsp;&nbsp;&nbsp; |  Dataset   &nbsp;&nbsp;&nbsp;&nbsp;&nbsp;&nbsp;&nbsp;&nbsp;   | GS&nbsp;&nbsp;&nbsp;&nbsp;&nbsp;&nbsp;&nbsp;| STDP-BW &nbsp;&nbsp;&nbsp;&nbsp;&nbsp;| STDP-BW-GS &nbsp; |    GSI  &nbsp;&nbsp;&nbsp;&nbsp;&nbsp;&nbsp;     |   GSGI  &nbsp;&nbsp;&nbsp;&nbsp;  |    Improvement |
> _____________________________________________________________________________________________________________
> | Spiking Resnet 18       |  DVS Gesture   | 93.14%  |     94.79%&nbsp;&nbsp;&nbsp;&nbsp;&nbsp;    |     95.14%&nbsp;&nbsp;&nbsp;&nbsp; &nbsp;&nbsp;&nbsp;&nbsp;&nbsp;&nbsp;        |   95.47 %&nbsp;&nbsp;&nbsp;&nbsp; |  95.83%&nbsp;&nbsp;&nbsp; |       2.69%        |
> _____________________________________________________________________________________________________________
> For Question3:
>
> The experiment you proposed only shows that the adjustment of the K value achieves an 8:2 ratio of excitatory and inhibitory mechanisms, which does not fully prove the effectiveness of the method. To this end, we added two sets of experiments to adjust the K value to 6 in the CIFAR10 and CIFAR100 data sets, that is, the ratio of excitation and inhibition mechanisms reached 10:2. The experimental results are shown in Table 4. From the experimental results, it can be seen that when K=6, the accuracy of GSI and GSGI training in the two data sets is lower than that of K=5. This also shows that the 8:2 ratio of excitation and inhibition mechanisms at K=5 is indeed the best ratio, which proves the correctness of our proposed ideas and improves biological interpretability.The table is shown below：
>
> Table4:Supplementary experiment: K=6(The ratio of excitatory mechanism to inhibitory mechanism is 10:2) on CIFAR10 and CIFAR100.
> ________________________________________________________________________________________
> SNN Model  &nbsp;&nbsp;&nbsp;&nbsp;&nbsp;&nbsp;&nbsp;&nbsp;&nbsp;                 Parameter K  &nbsp;&nbsp;&nbsp;&nbsp;&nbsp;&nbsp;&nbsp;         Learning Method &nbsp;&nbsp;&nbsp;&nbsp;     CIFAR10&nbsp;&nbsp;        CIFAR100
> ________________________________________________________________________________________
>   Spiking Resnet 18      &nbsp;&nbsp; &nbsp;&nbsp;&nbsp;        6        &nbsp;&nbsp;&nbsp;&nbsp;&nbsp;&nbsp;&nbsp;&nbsp;&nbsp;&nbsp;&nbsp;&nbsp;&nbsp;&nbsp;&nbsp;&nbsp;&nbsp;&nbsp;&nbsp;&nbsp; &nbsp;&nbsp;&nbsp;&nbsp;&nbsp; GSI  &nbsp;&nbsp;&nbsp;&nbsp;&nbsp; &nbsp;&nbsp;&nbsp;&nbsp;&nbsp; &nbsp;&nbsp;&nbsp;&nbsp;&nbsp;            89.65%. &nbsp;&nbsp;  &nbsp;&nbsp;&nbsp;&nbsp; &nbsp;      54.27%
>
>  Spiking Resnet 18     &nbsp;&nbsp;  &nbsp;&nbsp;&nbsp;         6    &nbsp;&nbsp;&nbsp;&nbsp;&nbsp;&nbsp;&nbsp;&nbsp;&nbsp;&nbsp;&nbsp;&nbsp;&nbsp;&nbsp;&nbsp;&nbsp;&nbsp;&nbsp;&nbsp;       &nbsp;&nbsp;&nbsp;&nbsp;             GSGI  &nbsp;&nbsp;&nbsp;&nbsp;&nbsp;&nbsp;&nbsp;&nbsp;&nbsp;&nbsp;&nbsp;&nbsp;&nbsp;&nbsp; &nbsp;&nbsp;                 90.34%&nbsp;&nbsp;&nbsp;&nbsp; &nbsp; &nbsp;&nbsp;      63.47%
> ________________________________________________________________________________________

---

> > ### Comment · Reviewer_5jTx · 2023-08-12
> > **second response**
> >
> > Thank you for your reply, which has solved my previous problem very well. I reviewed the article again, and I would like to know about the design of the iterative training of the excitatory mechanism and the inhibitory mechanism, please provide some explanations and descriptions.

---

> > > ### Author Response · Authors · 2023-08-13
> > > **Reply to Reviewer 5jTx**
> > >
> > > Thanks for your comment , we hope this clarifies further.
> > >
> > > Regarding the motivation of excitatory versus inhibitory iterative training, we explain as follows:
> > > Since the current training mechanism of SNNs is independent training, the core idea of this paper is to combine the excitatory mechanism with the inhibitory mechanism. At this time, how to train two different methods under the same network is very important. Here we think of using an iteraive training method with time T. Since the spiking neural network has time characteristics, we combine the time step T and the number of training rounds epoch to achieve the extension in the time dimension. The excitatory mechanism and the inhibitory mechanism are trained separately in two rounds of epoch, which is equivalent to training the excitatory mechanism in the 0-T time period in the time dimension, and the local maximum of the excitatory mechanism trained in the previous T time period in the T-2T time period. Value of merit, continue to train the inhibitory mechanism. In this way, the excitatory and inhibitory mechanisms can be trained synchronously in the same network and influence each other to achieve their respective optimal values. The combination in this case can test that the excitatory mechanism and the inhibitory mechanism can not only use their respective characteristics to achieve their own local optimum, but also use the characteristics of the two to promote each other, imitating the visual nervous system in biology. mechanism to achieve the optimal solution for joint training.

---

> > ### Comment · Reviewer_4jP2 · 2023-08-12
> >
> > Thanks for the authors' efforts in adding more experiments. The previous concerns can be solved to a great extent. It is better, if possible, to add more results under different K values for make the conclusion more convincing.

---

> > > ### Author Response · Authors · 2023-08-18
> > > **Reply to Reviewer 4jP2**
> > >
> > > Thanks for your comment , we hope this clarifies further.
> > >
> > > For your proposal of adding more results under different K values, we have added several sets of experiments to train the DVS Gesture dataset under the Spiking Resnet network model with different K values. The experimental results are as follows,
> > >
> > > ___________________________________________________
> > > K value    &nbsp;   &nbsp;&nbsp;&nbsp;&nbsp;&nbsp;          GSI     &nbsp;&nbsp;&nbsp;&nbsp;&nbsp;&nbsp;   GSGI
> > >
> > > &nbsp;&nbsp;&nbsp;2 &nbsp;&nbsp;&nbsp;&nbsp;&nbsp;&nbsp;&nbsp;&nbsp;&nbsp;&nbsp;  95.47%&nbsp;&nbsp;&nbsp;&nbsp;&nbsp;&nbsp; 95.83%
> > >
> > > &nbsp;&nbsp;&nbsp;3&nbsp;&nbsp;&nbsp;&nbsp;&nbsp;&nbsp;&nbsp;&nbsp;&nbsp;&nbsp;&nbsp;   95.49%&nbsp;&nbsp;&nbsp;&nbsp;&nbsp;&nbsp; 95.97%
> > >
> > > &nbsp;&nbsp;&nbsp;4&nbsp;&nbsp;&nbsp;&nbsp;&nbsp;&nbsp;&nbsp;&nbsp;&nbsp;&nbsp;&nbsp;  95.83%&nbsp;&nbsp;&nbsp;&nbsp;&nbsp;&nbsp;96.18%
> > >
> > > &nbsp;&nbsp;&nbsp;5&nbsp;&nbsp;&nbsp;&nbsp;&nbsp;&nbsp;&nbsp;&nbsp;&nbsp;&nbsp;&nbsp;   96.18%&nbsp;&nbsp;&nbsp;&nbsp;&nbsp;&nbsp;96.53%
> > >
> > > &nbsp;&nbsp;&nbsp;6&nbsp;&nbsp;&nbsp;&nbsp;&nbsp;&nbsp;&nbsp;&nbsp;&nbsp;&nbsp;&nbsp;  95.49%&nbsp;&nbsp;&nbsp;&nbsp;&nbsp;&nbsp;95.83%
> > > __________________________________________________
> > > From the experimental results, it can be seen that as the K value increases, the accuracy rate also increases. When K=5, that is, when the ratio of excitatory mechanism to inhibitory mechanism reaches 4:1, the accuracy rate reaches its peak. Make our ideas more convincing.

---

> > > > ### Comment · Reviewer_5jTx · 2023-08-21
> > > > **Thanks for your response**
> > > >
> > > > I thank the authors for answering my questions. The authors address my concern well. Besides, the authors illustrated the role how different k plays in the EICIL and discuss the detailed ratio of excitatory mechanism to inhibitory mechanism. How this mechanism was bridged between neuroscience and AI is meaningful, I would like to see the authors make more efforts in the future.

---

### Official Review · Reviewer_a2bZ · 2023-06-29

**Soundness:** 3 good
**Presentation:** 2 fair
**Contribution:** 2 fair
**Rating:** 8
**Confidence:** 4

**Summary:**

The paper is about a new learning method for spiking neural networks (SNNs). The authors introduce a method called EICIL that integrates both excitatory and inhibitory behaviors into one network, which improves plausibility and adaptability. They show that EICIL performs better than traditional methods on various datasets, such as CIFAR10 and CIFAR100.

**Strengths:**

The paper has several strengths:
1. The number of people paying attention to the STDP algorithm is dwindling at NeurIPS conferences. This article provides a good idea to merge STDP with the common backpropagation methods of deep spiking neural networks, so that STDP can help the network learn.

2. The process of integrating STDP is actually not complicated. The method first classifies the neurons (excitatory neurons or inhibitory neurons) and adjusts the synaptic connections according to the classes of neurons.

3. This method works especially well on the CIFAR100 dataset.

**Weaknesses:**

The presentation of the paper needs improvement:
1. Why can "Surrogate Gradient Method" be abbreviated as "GS", not "SG"? It is confusing.
2. There should be a period or comma after each equation.
3. In the term "Excitatory inhibition Net", the rule of capitalization is too causal. I think the authors should also capitalize the word "inhibition". Also "Resnet" should be "ResNet".
4. There are places missing blanks, e.g., lines 42, 103, 147, and 173.
5. In Section 5.4, k should be "$k$".
6. There are multiple expressions of the CIFAR-10 dataset: "CIFAR10", "CIFAR-10" , "CIFAR 10". Please make them consistent.
7. Figure (a) is too small to read.

**Questions:**

From Lines 210–222, I get the knowledge that the cycle iteration learning is to let excitatory and inhibitory circuits take turns to learn. However, I can only see one formula (equation 9) that generates updates. Can you further explain this to me?



What is "BN" in the paper? I can only see the abbreviation in the paper.

**Limitations:**

The presentation of this paper can be improved.

---

> ### Author Rebuttal · Authors · 2023-08-08
>
> Thank you very much for your careful review and constructive comments, which have much contributed to improving the paper. We will address all the comments in the paper and provide responses to the comments below.
>
> We have made revisions for writing problems
>
> For Weakness1:
>
> The reason why the gradient substitution method is abbreviated as GS here is to make it easier to distinguish the naming after combining the gradient substitution method with the STDP method in method4.3.2. G in GSI stands for gradient substitution, S stands for STDP, and I stands for iteration. It would be confusing if it was abbreviated as SG.
>
> For Question1:
>
> For the update question you raised:Equation 5 describes that in the STDP method, the weights are updated according to the time difference between the spike excitations between adjacent neurons
>
> \begin{equation}\label{Eq:method1}
> \frac{dV_j(t)}{dt} = \sum_{i=1}^N w_{ij} F(s_j(t-\Delta)-\theta_j) + \sum_{i=1}^N w_{ij} G(s_j(t-\Delta)-s_i(t-\Delta)-\theta_{ij})
> \end{equation}
> where $w_{ij}$ represents the connection weight between neurons $i$ and $j$, $\theta_j$ represents the threshold of neuron $j$, $s_j(t)$ represents the signal indicating whether neuron $j$ has fired at time $t$, and $\Delta$ represents the time delay of spiking propagation. And this equation expresses the rate of change in the membrane potential of a neuron in response to input from other neurons, representing its excitatory or inhibitory nature.
>
> \begin{equation}
> W_{t+1} = W_t - \eta \frac{\partial W_t}{\partial L}
> \end{equation}
> \begin{equation}
> \Delta W_{ij}(t) = \sum_{k \in S^+} f_+(t-t_k) - \sum_{k \in S^-} f_-(t-t_k)
> \end{equation}
> \begin{equation}
> W_{t+1} = W_t + \eta \Delta W_t
> \end{equation}
>
> Equation 8 and Equation 9 indicate that the two combine their own weight update characteristics to jointly update the weight parameters, especially in Equation 9, which reflects the excitatory and inhibitory spike sequences, as well as the spike spatiotemporal sequence. The combined update weights in Equation 10 combine the joint update of excitation and inhibition with separate updates of excitation and inhibition. From the perspective of biological science, the excitatory mechanism and the inhibitory mechanism have their own system structure to maintain their independent activities. But when the two act on neurons together, they use the mechanism that jointly acts on neurons to affect neurons to generate spike.
>
>
>
> For Question2:
>
> The BN layer here refers to the Batch Normalization layer. It is also a layer of the network like the convolutional layer, pooling layer, and fully connected layer.

---

> > ### Comment · Reviewer_a2bZ · 2023-08-11
> > **Reply to the authors**
> >
> > Thank you for your reply, which has solved my previous problem very well. Now that I've reviewed the article again, I wonder why increasing or decreasing K in Table 2 causes performance to go up for some data sets and methods and down for others. Please provide some insight. If you answer well, I will give you extra points.

---

> > > ### Author Response · Authors · 2023-08-13
> > > **Reply to Reviewer a2bZ**
> > >
> > > Thanks for your comment , we hope this clarifies further.
> > >
> > > For your question, we have the following reply:
> > > First of all, we have the following explanation for the setting of the K value in Table 2: the ratio of excitatory and inhibitory mechanisms in the visual nervous system in biological brains in biological sciences is approximately 4:1[1][2]. Combined with the idea of combining the excitatory mechanism and the inhibitory mechanism proposed in this paper, we think that the time ratio of the current iterative training is to train the excitatory mechanism during the T period, and to train the inhibitory mechanism during the T period, while the spiking neural network and even the artificial neural network model They are all constructed by imitating the neural network in biology. Therefore, we consider that if the ratio of excitatory mechanism and inhibitory mechanism is gradually increased to gradually reach the ideal ratio of 4:1 in biological sciences, whether the final model training effect will achieve the best accuracy at the 4:1 moment. After experimental verification, using the GSI method or GSGI method for iterative training on the CIFAR10 and CIFAR100 data sets, the final results are all in the direction of 1:1 to 4:1 growth, and the accuracy rate is indeed gradually improving. However, in the supplementary experiment (Table 4), when the combination ratio of excitatory and inhibitory mechanisms is higher than 4:1, both training methods show a clear downward trend on both datasets, thus proving our hypothesis: in The best training effect can be achieved when the ratio of the combination of excitatory mechanism and inhibitory mechanism in the spiking neural network is close to the ideal ratio of 4:1 in the visual nervous system in biological sciences. The proposal of this idea also demonstrates that our idea enhances the biological interpretability of spiking neural networks. As for the declining results in the MNIST data set, we analyzed that the training results in MNIST have reached a data set with an accuracy rate close to 100%, and the range of improvement itself is limited, so the effectiveness of this method has not been shown for the time being. But there is a clear improvement in other datasets where the training accuracy is not close to the maximum.
> > >
> > > [1]Jie Zhu,Man Jiang,Mingpo Yang,Han Hou,Yousheng Shu. Membrane Potential-Dependent Modulation of Recurrent Inhibition in Rat Neocortex Published: March 22, 2011
> > >
> > > [2]Robin Meadows. Finding Balance in Cortical Networks Published: March 22, 2011

---

> > > > ### Comment · Reviewer_a2bZ · 2023-08-16
> > > > **Thanks for your clarification**
> > > >
> > > > Thanks for your clarification. My questions have been successfully solved. Hence, I would like to increase my socre.

---

### Official Review · Reviewer_4jP2 · 2023-07-02

**Soundness:** 3 good
**Presentation:** 3 good
**Contribution:** 3 good
**Rating:** 6
**Confidence:** 4

**Summary:**

The article introduces a training method of spiking neural network, which combines the excitatory mechanism with the inhibitory mechanism. Through iterative training, the same network model is jointly trained at different time steps. The article fully combines the excitatory and inhibitory properties of the neural network structure in the biological brain with the artificial neural network.

**Strengths:**

1. The article combines the excitatory mechanism with the inhibitory mechanism, imitating the neural network mechanism in the biological brain, and enhancing the biological interpretability of the artificial neural network.
2. The article makes full use of the unique hyperparameters of the spiking neural network: the time step to combine the excitatory mechanism with the inhibitory mechanism, allowing the two to jointly train the same network structure, and using each other for local optimal iterative training. This is an innovative breakthrough.
3. Combining the excitatory-inhibitory ratio of the biological visual nervous system, the article sets the excitatory mechanism and the inhibitory mechanism in the spiking neural network to an ideal ratio of 8:2. It is verified by experiments that the accuracy rate obtained by this ratio is higher than the original 1:1. As a result, the effectiveness of the method is proved.

**Weaknesses:**

1. The expression of the article is not clear enough, and the authors’ idea is not expressed with the most concise sentence.
2. The article introduces just one combination of an excitatory mechanism and an inhibitory mechanism. If there are more combinations to prove the idea proposed by the authors, it will be more convincing
3. The methods used in the article are all the most basic methods, which have not yet reached the current highest accuracy rate. Although the comparison experiment has improved to a certain extent, it is still unknown whether it can improve the current highest accuracy rate.

**Questions:**

1. The data sets used in the experimental part are relatively traditional image data. I don’t know whether this method will also be applicable to pulse-form data sets, and there will be a significant improvement.
2. The article only introduces a combination of excitation and inhibition. If another combination of excitation and inhibition methods can be added, it will further prove the feasibility of the method.
3. The K value proposed in the article adjusts the ratio of the excitatory mechanism over the inhibitory mechanism. The experiments only showed a ratio of 8:2. It is not known whether the result is the best ratio. If an excitatory mechanism is increased with a ratio greater than 8:2, it will be better for the proof of the method.

**Limitations:**

The authors have not explicitly discussed limitations of their approach. Since this work is theoretical, it is unlikely it will have potential negative societal impacts.

---

> ### Author Rebuttal · Authors · 2023-08-08
>
> Thank you very much for your careful review and constructive comments, which have much contributed to improving the paper. We will address all the comments in the paper and provide responses to the comments below.
>
> We have made revisions for writing problems
>
> For Question1:
>
> As for whether our method proposed by you can be applied to other more complex data sets, we have added a set of experiments. The experimental data set is the DVSGesture data set. The "DVSGesture" data set is a gesture recognition data captured based on dynamic visual sensors. set. The main feature of the DVSGesture dataset is that it contains impulsive event sequences of gesture motions captured on dynamic vision sensors. Each pulse event records a change in pixel value within the region of interest, usually expressed as a timestamp, pixel coordinate, and polarity (increase or decrease).Please refer to Table 3(See the attachment or the table below for details)  for the experimental results. The experimental results show that the STDP-BW-GS method reaches 95.14\%, which is higher than the 94.79\% of the STDP-BW method, which confirms the idea of method 4.3.1. At the same time, the results of GSGI and GSI are both higher than the accuracy of the GS method baseline by at least 2\%, which proves the effectiveness of the idea in method4.3.2. And the experimental results also show that the result of GSGI is higher than that of GSI, which also proves that the STDP-BW-GS method is better than STDP-BW. Therefore, experiments have proved that our method is also applicable to more complex data sets, and the robustness and generalization of the method have been verified.The table is shown below：
>                                               Table3:Test accuracies of EICIL on DVS Gesture.
> _____________________________________________________________________________________________________________
> | SNN Model &nbsp;&nbsp;&nbsp;&nbsp;&nbsp;&nbsp;&nbsp;&nbsp;&nbsp;&nbsp;&nbsp; |  Dataset   &nbsp;&nbsp;&nbsp;&nbsp;&nbsp;&nbsp;&nbsp;&nbsp;   | GS&nbsp;&nbsp;&nbsp;&nbsp;&nbsp;&nbsp;&nbsp;| STDP-BW &nbsp;&nbsp;&nbsp;&nbsp;&nbsp;| STDP-BW-GS &nbsp; |    GSI  &nbsp;&nbsp;&nbsp;&nbsp;&nbsp;&nbsp;     |   GSGI  &nbsp;&nbsp;&nbsp;&nbsp;  |    Improvement |
> _____________________________________________________________________________________________________________
> | Spiking Resnet 18       |  DVS Gesture   | 93.14%  |     94.79%&nbsp;&nbsp;&nbsp;&nbsp;&nbsp;    |     95.14%&nbsp;&nbsp;&nbsp;&nbsp; &nbsp;&nbsp;&nbsp;&nbsp;&nbsp;&nbsp;        |   95.47 %&nbsp;&nbsp;&nbsp;&nbsp; |  95.83%&nbsp;&nbsp;&nbsp; |       2.69%        |
> _____________________________________________________________________________________________________________
>
> For Question2:
>
> For the applicability of another combination, the iterative training method that we propose combines excitation and inhibition is aimed at training the same network with the excitatory mechanism and the inhibitory mechanism. In theory, as long as the two methods can be used in the same network If the training is carried out in the middle, then no matter which training method is selected as the excitatory mechanism or the inhibitory mechanism, as long as one method is the excitatory mechanism and the other is the inhibitory mechanism, then the method is effective. At the same time, we consider that the two existing methods are sufficiently representative in the training methods of spiking neural networks, and due to equipment limitations, there is not enough time for experiments. But in the future, we will explore what method is more suitable for our method, and continue to explore how to optimize the method and improve the efficiency and accuracy of the combination of the two.
>
> For Question3:
>
> The experiment you proposed only shows that the adjustment of the K value achieves an 8:2 ratio of excitatory and inhibitory mechanisms, which does not fully prove the effectiveness of the method. To this end, we added two sets of experiments to adjust the K value to 6 in the CIFAR10 and CIFAR100 data sets, that is, the ratio of excitation and inhibition mechanisms reached 10:2. The experimental results are shown in Table 4. From the experimental results, it can be seen that when K=6, the accuracy of GSI and GSGI training in the two data sets is lower than that of K=5. This also shows that the 8:2 ratio of excitation and inhibition mechanisms at K=5 is indeed the best ratio, which proves the correctness of our proposed ideas and improves biological interpretability.The table is shown below：
>
> Table4:Supplementary experiment: K=6(The ratio of excitatory mechanism to inhibitory mechanism is 10:2) on CIFAR10 and CIFAR100.
> ________________________________________________________________________________________
> SNN Model  &nbsp;&nbsp;&nbsp;&nbsp;&nbsp;&nbsp;&nbsp;&nbsp;&nbsp;                 Parameter K  &nbsp;&nbsp;&nbsp;&nbsp;&nbsp;&nbsp;&nbsp;         Learning Method &nbsp;&nbsp;&nbsp;&nbsp;     CIFAR10&nbsp;&nbsp;        CIFAR100
> ________________________________________________________________________________________
>   Spiking Resnet 18      &nbsp;&nbsp; &nbsp;&nbsp;&nbsp;        6        &nbsp;&nbsp;&nbsp;&nbsp;&nbsp;&nbsp;&nbsp;&nbsp;&nbsp;&nbsp;&nbsp;&nbsp;&nbsp;&nbsp;&nbsp;&nbsp;&nbsp;&nbsp;&nbsp;&nbsp; &nbsp;&nbsp;&nbsp;&nbsp;&nbsp; GSI  &nbsp;&nbsp;&nbsp;&nbsp;&nbsp; &nbsp;&nbsp;&nbsp;&nbsp;&nbsp; &nbsp;&nbsp;&nbsp;&nbsp;&nbsp;            89.65%. &nbsp;&nbsp;  &nbsp;&nbsp;&nbsp;&nbsp; &nbsp;      54.27%
>
>  Spiking Resnet 18     &nbsp;&nbsp;  &nbsp;&nbsp;&nbsp;         6    &nbsp;&nbsp;&nbsp;&nbsp;&nbsp;&nbsp;&nbsp;&nbsp;&nbsp;&nbsp;&nbsp;&nbsp;&nbsp;&nbsp;&nbsp;&nbsp;&nbsp;&nbsp;&nbsp;       &nbsp;&nbsp;&nbsp;&nbsp;             GSGI  &nbsp;&nbsp;&nbsp;&nbsp;&nbsp;&nbsp;&nbsp;&nbsp;&nbsp;&nbsp;&nbsp;&nbsp;&nbsp;&nbsp; &nbsp;&nbsp;                 90.34%&nbsp;&nbsp;&nbsp;&nbsp; &nbsp; &nbsp;&nbsp;      63.47%
> ________________________________________________________________________________________

---

### Official Review · Reviewer_imxR · 2023-07-26

**Soundness:** 2 fair
**Presentation:** 2 fair
**Contribution:** 2 fair
**Rating:** 3
**Confidence:** 3

**Summary:**

The paper presents the EICIL approach, an iterative excitatory/inhibitory learning mechanism to train task-specific spiking neural networks which still maintain a level of biological realism. The approach utilizes spike-timing dependent plasticity (STDP) as part of the inhibitory mechanism, showing that it can be scaled to multiple layers. The authors show promising results on the MNIST, CIFAR-10, and CIFAR-100 datasets, with slightly favorable results compared to a standard surrogate gradients approach. The authors also perform a simple ablation study by varying the number of excitatory/inhibitory iterations and show that this can influence final model performance.

**Strengths:**

The paper tackles an interesting problem, specifically biologically plausible neural networks based on spikes as the fundamental unit of computation. The authors propose an extension of existing spiking neural network methods by combining a tunable iterative excitatory/inhibitory learning mechanism. They show that their approach can be used with relatively standard convolutional neural network architectures, such as ResNet and VGG. The authors also performed detailed experiments across models and datasets, showing that their approach performs slightly favorably compared with other baselines.

**Weaknesses:**

Overall, the writing in the paper could be improved and the motivation/justification for certain design decisions made more clear. There were a couple of typos and incomplete sentences (e.g. line 314, "but due to the model structure and complexity" and line 330, "On the CIFAR100 data"). The motivation for using surrogate gradients for convolutional layers as the excitatory mechanism and STDP for the final linear layer as the inhibitory mechanism was somewhat unclear and seemed ad-hoc. Some of the connections to biological constraints were pretty forced (e.g. the 4:1 excitatory to inhibitory ratio and relating that to training ratio of excitatory/inhibitory mechanisms).

In general, I also found it hard to keep track of the different model configurations and their acronyms- I believe a figure showing the model architecture(s) would help the reader better understand the overall concept and map the different configurations to rows in the results tables. The experimental results were also somewhat weak, showing performance improvements that may not be statistically significant and could be purely due to randomness (e.g. <1%). The paper also claims that the EICIL approach "enhances biological interpretability" and "expands its representation space", but I do not really see this quantified anywhere in the paper. Are the representations learned by EICIL actually more biologically realistic or interpretable and how could this be demonstrated?

**Questions:**

1. Could you explain the design rationale for separating out the surrogate gradient and STDP excitatory/inhibitory learning mechanisms?

2. Could you please add a figure or diagram that shows the different model configurations (e.g. STDP-BW, STDP-BW-GS, GSI, GSGI)? It was sometimes hard to keep track of these purely based on descriptions in the text, and I believe such a figure would help the reader better understand the overall concept.

3. How would this approach scale to more complex datasets beyond MNIST and CIFAR? I am not very familiar with the current literature on spiking neural networks and associated benchmarks, but a discussion at least of how the proposed approach might scale to larger datasets would be useful.

**Limitations:**

I don't think there are many potential negative societal impacts of the current work, given its current framing and context. The authors do briefly address some particular limitations of their approach.

---

> ### Author Rebuttal · Authors · 2023-08-08
>
> Thank you very much for your careful review and constructive comments, which have much contributed to improving the paper. We will address all the comments in the paper and provide responses to the comments below.
>
> We have made revisions for writing problems
>
> For Weakness1:
> For the motivation of using GS for convolutional layers as the excitatory mechanism and STDP for the linear layer as the inhibitory mechanism we have the following explanations:the method of combining the excitatory and inhibitory mechanism acts on the overall network layer. That is, in the same epoch, we train the excitatory and inhibitory mechanisms on different layers. It can be understood that different layers in the same epoch training have different methods of updating weights. And the final linear layer is not mandatory to use STDP to train, but expresses a process of exploring the combination of excitatory and inhibitory mechanisms. The purpose is to observe whether this result is more in line with the expression of excitation and inhibition in biological neural networks. Facts have also proved that the effect is not ideal.The idea of using the convolutional layer to train the inhibitory mechanism and using other layers to train the excitatory mechanism is that the two are interleaved and promote each other, which is more in line with the expression of biological neural networks.And the experiment also proves that the result is indeed slightly better than GS.Similarly, these connections with biological constraints are not mandatory.It has been proved by experiments that the combination of excitatory and inhibitory mechanism is higher than other values when the ratio of excitatory and inhibitory mechanism reaches 4:1. And the 4:1 ratio is indeed a peak. This also proves that the setting of our model is consistent with the proportional distribution of excitatory and inhibitory mechanisms in biological neural networks.To explain with reference to biological science:There is a major interplay between excitation and inhibition. We considered a model of the primary visual cortex as a network consisting of excitatory and inhibitory cell populations, with both short and long range interactions[1] .So it shows that our proposed method enhances the biological interpretability of SNNs.
>
> For Weakness2:
> In view of whether there is randomness in the experimental results, we added a set of experiments: use GSI to train the CIFAR10 dataset under the SpikingResnet network model (Table 1). It proves that this method will improve every time, not a random improvement.And the main purpose of this paper is to illustrate this new training method that iteratively combines excitatory and inhibitory mechanisms in SNNs, not focusing on achieving a significant improvement effect. Of course, in the future, we will explore how other different and more advanced methods can improve SNNs.
> The biological interpretability problem stated in weakness1
> The expanding the representation space question, stated in Question3
>
> For Question1:
> The current GS or STDP methods use their own training features to update the weights independently. Although both have excitatory or inhibitory effects, there is no training method combining the two. In biological sciences, the excitatory and inhibitory mechanism are often interrelated and work together with neurons to transmit information. Therefore, this paper combines the excitatory and inhibitory mechanism in SNNs to make it more biologically interpretable. Since GS is widely used in SNNs, GS was used as the excitatory mechanism.  STDP is to update the weight parameters according to the spike generation time difference between two adjacent neurons. When multiple synapses are connected to the same neuron, if the strength of one synapse is enhanced, it may increase the activity threshold of subsequent synapses, causing the strength of these subsequent synapses to be suppressed.So it is also known as the side-suppression method. So we use STDP as an inhibitory mechanism. The idea of the combination of excitation and inhibition also broadens new ideas for the subsequent development of SNNs.
>
> For Question2:
> We have added an explanation of the different model in the Table2 . STDP-BW is the abbreviation of the method combining STDP with backpropagation in method4.2. STDP-BW-GS is the abbreviation of the method in the method4.3.1 that trains the STDP-BW method and GS separately on different network layers in the same network model. GSI and GSGI are the abbreviations of GS in method 4.3.2 to iteratively train the same network model with STDP-BW and STDP-BW-GS respectively. Display of some hyperparameters: the learning rate is selected as lr=0.002, the time step is selected as T=8, and the number of training rounds is epoch=320.
>
> For Question3:
> As for whether our method can be applied to other more complex data sets, we have added a set of experiments. The experimental data set is the DVSGesture dynamic dataset.  Please refer to Table 3 for the experimental results. The experimental results show that the STDP-BW-GS reaches 95.14%, which is higher than the 94.79% of the STDP-BW, which confirms the idea of method4.3.1. The results of GSGI and GSI are both higher than the accuracy of GS by at least 2%, which proves the effectiveness of the idea in method4.3.2. And the experimental results also show that the result of GSGI is higher than that of GSI, which also proves the idea of method4.3.1.Therefore, experiments have proved that our method is also applicable to more complex data sets, and the robustness and generalization of the method have been verified. At the same time, it shows that this method not only has an improvement in static data sets, but also has an excellent improvement effect when extended to dynamic data sets. The expansion effect on the representation space is proved.
>
> [1]Yael Adini, Dov Sagi, and Misha Tsodyks .Excitatory–inhibitory network in the visual cortex: Psychophysical evidence.September 16, 1997

---

> > ### Comment · Reviewer_imxR · 2023-08-11
> >
> > I have read the author's rebuttal. Unfortunately, I am still concerned about the presentation of results. For example, It is not entirely clear to me what value Table 1 in the rebuttal adds- it shows performance on the CIFAR 10 dataset across 10 seeds(?), yet the difference in performance with the baseline GS method is still <1%. I am also still uncertain about whether the proposed work actually improves the biological interpretability and representation learning of SNNs, which is one of the authors' claims. I feel these are still not entirely quantified, and many of the design choices in EICIL still seem relatively arbitrary and ad-hoc to me. As a result, I still maintain my original rating.

---

> > > ### Author Response · Authors · 2023-08-13
> > > **Reply to Reviewer imxR**
> > >
> > > Thanks for your  comment , we hope this clarifies further.
> > >
> > > For your question, we have the following reply:
> > > First of all, the 10 sets of data we added were run from different seeds, which already shows that our experiment is not random, but indeed improved. At the same time, the improvement of the VGG network in CIFAR10 and CIFAR100 is greater than 1%, and although the CIFAR10 data set in the Res net network has not increased to 1%, it is already close to 1%. This also shows that our experiment is not random.Secondly, regarding the question of improving biological interpretability and representation space you mentioned, combined with the examples we mentioned in rebuttal, as well as the literature in biological science [1][2] combined as follows, we believe is sufficient to show that the realization of our idea indeed improves the biological interpretability of spiking neural networks.Because the realization of our ideas does make the original biologically interpretable spiking neural network more in line with the characteristics of the nervous system in biological sciences,and it is proposed that the excitatory mechanism and inhibitory mechanism in the spiking neural network reach a ratio of 4:1, which is closer to the ratio of excitation and inhibition in the visual nervous system of the brain.These are enough to prove that our idea improves the biological interpretability of spiking neural networks. At the same time, for the improvement of the representation space, our method is not only applicable to static data sets such as CIFAR data sets and MNIST data sets, but also to dynamic data sets such as DVS Gesture (see Supplementary Experiment Table 3),which shows that it broadens the representation space of artificial neural networks that can only be applied to static data sets.Finally, many of the designs in the article are verified through experiments, not mandatory settings, and forced explanations. Each part of the article can be explained by corresponding biological science and computer science, and we have all experimented to verify our ideas. So it is not forced to set, deliberately explained.
> > >
> > > [1]Jie Zhu,Man Jiang,Mingpo Yang,Han Hou,Yousheng Shu. Membrane Potential-Dependent Modulation of Recurrent Inhibition in Rat Neocortex Published: March 22, 2011
> > >
> > > [2]Robin Meadows. Finding Balance in Cortical Networks Published: March 22, 2011

---

> > > ### Author Response · Authors · 2023-08-21
> > > **Reply to Reviewer imxR**
> > >
> > > Dear Reviewer, Thank you so much for your review -- please let us know if you have any remaining questions or concerns so that we can address them before the deadline coming soon. Alternatively, if you feel that your original concerns are addressed, we would appreciate updating your evaluation to reflect that. Thank you!

---

### Author Rebuttal · Authors · 2023-08-08

We will revise the final version. For better presentation, Reviewer imxR, Reviewer 4jP2, Reviewer a2bZ and Reviewer 5jTx were noted as R1, R2, R3 and R4.

We sincerely thank all reviewers for their time and thoughtful feedback. In particular, we are grateful for the largely positive reception of our work:

The reviewers seem to agree with our proposal for an innovative training method for spiking neural networks[R1,R2,R3,R4]“The authors propose an extension of existing spiking neural network methods by combining a tunable iterative excitatory/inhibitory learning mechanism.” [R1] “a good idea”[R3]“an innovative breakthrough ”[R2，R4].We will incorporate all feedback and correct all typos and incorporate suggestions.

Contributions:

This paper combines the excitatory mechanism with the inhibitory mechanism, imitating the neural network mechanism in the biological brain, and enhancing the biological interpretability of the artificial neural network.

This paper makes full use of the unique hyperparameters of the spiking neural network: the time step to combine the excitatory mechanism with the inhibitory mechanism, allowing the two to jointly train the same network structure, and using each other for local optimal iterative training.

This paper also designs the ideal ratio of 8:2 of excitatory and inhibitory mechanisms to the neural network model in biological brains, making the above method more persuasive.

This paper improves the training mode of STDP, so that STDP can help the network to learn. And it improves the accuracy of STDP training multi-layer neural network.

The reviewers raised individual points of criticism, which we address in detail in our direct responses to each reviewer. The main comment is to add a more complex dataset to test our ideas. Here we have added a set of experiments on the DVS Gesture dataset, successfully verifying that our method is also adaptable to more complex datasets.We also fully illustrate and explain how the method improves biological interpretability.In addition, we also proposed to explain and verify the idea that the combination ratio of excitatory mechanism and inhibitory mechanism here reaches 8:2.In response to these questions you raised, we will fully explain and respond below, and supplemented several sets of related experiments (see Table 1-4) to fully verify the feasibility of this idea.

We believe that the remarks of the reviewers and the additional results further improved the paper, and will happily respond to any additional questions or comments during the discussion phase.

---

### Decision · Program_Chairs · 2023-09-21

**Decision:**

Accept (poster)

**Comment:**

The reviewers mentioned several positive aspects of the paper, e.g.:
- The paper proposes an extension of existing spiking neural network methods by combining a tunable iterative excitatory/inhibitory learning mechanism.
- The article combines the excitatory mechanism with the inhibitory mechanism, imitating the neural network mechanism in the biological brain, and enhancing the biological interpretability of the artificial neural network.
-  This article provides a good idea to merge STDP with the common backpropagation methods of deep spiking neural networks so that STDP can help the network learn.
- This method works especially well on the CIFAR100 dataset.

However, they also raised multiple doubts about it, such as:
- The writing in the paper could be improved and the motivation/justification for certain design decisions made more clear.
- It is hard to keep track of the different model configurations and their acronyms.
- The paper only introduces the combination of one excitatory mechanism and one inhibitory mechanism and does not fully express the effectiveness and feasibility of the method.

Three out of four reviewers agree (8-7-6 vs. 3) that the paper should be accepted. I tend to agree with them, thus, I vote to accept the paper.